# Rural–Urban Transition of Hanoi (Vietnam): Using Landsat Imagery to Map Its Recent Peri-Urbanization

**Giovanni Mauro** 

Department of Humanities, University of Trieste, via Lazzaretto Vecchio 8, 34100 Trieste, Italy; gmauro@units.it;
Tel.: +(39)-040-5583631

**Abstract:** The current trend towards global urbanization presents new environmental and social challenges. For this reason, it is increasingly important to monitor urban growth, mainly in those regions undergoing the fastest urbanization, such as Southeast Asia. Hanoi (Vietnam) is a rapidly growing medium-sized city: since new economic policies were introduced in 1986, this area has experienced a rapid demographic rise and radical socio-economic transformation. In this study, we aim to map not only the recent urban expansion of Hanoi, but also of its surroundings. For this reason, our study area consists of the districts within a 30km radius of the city center. To analyze the rural–urban dynamics, we identified three hypothetical rings from the center: the core (within a 10 km radius), the first ring (the area between 10 and 20 km) and, finally, the outer zone (over 20 km). To map land use/land cover (LULC) changes, we classified a miniseries of Landsat images, collected approximately every ten years (1989, 2000, 2010 and 2019). To better define the urban dynamics, we then applied the following spatial indexes: the rate of urban expansion, four landscape metrics (the number of patches, the edge length, the mean patch area and the largest patch index) and the landscape expansion index. The results show how much the city's original shape has changed over the last thirty years: confined for hundreds of years in a limited space on the right bank of the Red River, it is now a fringed city which has developed beyond the river into the surrounding periurban areas. Moreover, the region around Hanoi is no longer solely rural: in just thirty years, urbanization has converted this territory into an industrial and commercial region.

**Keywords:** rural–urban transition; Landsat imagery; LULC; periurbanization; Hanoi

## 1. Introduction

In the last few decades, the remarkable increase in the number of people living in urban areas around the world has posed new problems and created fresh challenges for urban planning and urban policies. More than four billion people (about 55% of the world's population) are now living in cities and it is likely that all of these regions will be even more urbanized in the coming years [1]. Even though, in 2014, close to one half of the world's urban population was living in settlements with fewer than 500,000 inhabitants, the trends indicate that the number of medium-sized cities (of 1–5 million people) and large cities (of 5–10 million people) located in Africa and in Asia is growing rapidly, often by the expansion of slums [2].

In 1990, there were ten cities in the world with more than 10 million inhabitants: in 2018, the number had tripled to 33, 19 of which are in Asia [3], often through a combination of recent economic growth and high-speed rural–urban transition. One of the fastest urbanizing regions in the world is East and Southeast Asia, which has seen urban land expansion at unprecedented rates in towns with increasing density [4].

Although it is still a predominantly rural country (agricultural and forestry land accounting for almost 80% of land use in 2019), the people living in urban areas of Vietnam account for about 35%

of the population (about 96.5 million in 2019). In a bipolar territorial layout, the city dwellers are mostly concentrated in the country's two main metropolitan regions: Hanoi (4 million) in the north and Ho Chi Minh City (7.2 million) in the south [5]. In the last three decades, both of these cities have experienced rapid demographic growth, mainly due to the radical socioeconomic transformation taking place as a result of a program of economic reforms known as Doi Moi ('renovation'), officially adopted in 1986. As shown by [6], since the early 1990s, economic development has given rise to rapid urbanization, improving social conditions but often causing environmental degradation. In contrast to the collectivist period (1975–1986), characterized by an anti-urban policy [7], cities are now regarded as the new key drivers of economic growth [8]. Today, Hanoi and Ho Chi Minh City are the key regions of contemporary Vietnam, ensuring nearly 70% of the Gross Domestic Product (GDP) in the service and manufacturing sectors [9].

Located to the north of the Red River delta region, Hanoi celebrated 1000 years of its troubled history in 2010. In view of the changes that have taken place in the Doi Moi period, several studies have explored the series of new challenges Hanoi is facing: the re-organization of its urban transportation system, i.e., [10,11]; the protection of its cultural heritage, i.e., [10–12], whose importance has recently been recognized by UNESCO (since 2010, the World Heritage List has included the Central Sector of the Imperial Citadel of Thang Long); how to maintain and expand its economic role within Vietnam, mainly in comparison to Ho Chi Minh City, i.e., [13,14].

The rapid urbanization of Hanoi, which has taken place at an unprecedented pace over the last few decades, makes it a good example of a specific model of Asian urbanization, namely periurbanization. In rapidly developing Asian countries such as China and Vietnam, the fast urban growth in recent years has created what are called extended metropolitan regions [15], namely urban areas usually surrounded by densely populated rural areas. As is well known, when a city grows, it spills out into the countryside. In North America, as in Western Europe, this phenomenon is referred to as urban sprawl. In Asian countries, the periurban areas are "where urban and rural processes meet, mix and interact" [16] (p. 1147), "a zone of encounter, conflict and transformation surrounding large cities" [17] (p. 426), where it is important "to distinguish between the near and the far periurban in relation to the central city" [18] (p. 164). This Asian spatial model, named desakota (a term derived from the Indonesian words for village (des) and city (kota)) [15], usually encompasses an area included between the inner core city and the outer zone (the "near periurban"). Here, the urban fringes grow very quickly beyond the limits of the city, often wiping out the pre-existing villages. Beyond this first zone, there is the "far periurban", an area which is often very fragmented, and characterized locally by a high population density, small-holder agriculture (mainly rice crop) and a well-developed infrastructure of roads and canals. This area is very attractive for urban activities: the lower labor costs have encouraged industrial relocation in these well-connected rural areas, located near (but outside) the city where the environmental pressures (water and atmosphere pollution) and social issues (i.e., poverty and housing) now represent relevant problems. As well known, in the urban metabolism model, "a city can be defined as all the materials and commodities needed to sustain the city's inhabitants at home, at work and at play" [19] (p. 176). For these reasons, following this perspective (i.e., [19,20]), the periurban areas could be re-imagined as "non-city zones", where "a range of metabolic byproducts (waste, pollution, carbon) are produced within and, eventually, absorbed back" [21] (p. 25). If Hanoi city—with its very high density (more than 2400 persons/Km$^2$ [5])—can represent the concept of concentrated urbanization well, its surroundings could be assimilated into operational landscapes: "non-city space transformed from capital forms of urbanization into zones of high-intensity, large-scale industrial infrastructure" [22] (p. 125).

Today, the methodology of remote sensing is frequently used to map land cover or, from a diachronic perspective, the changes in land cover occurring in recent times at several different scales (local, regional, etc.). Since 2008, the United States Geological Survey (USGS) has made its Landsat archive freely available [23]. This database of medium-scale satellite images covers more than 50 years and provides us with free, detailed and consistent geographical information; thus, interest in the use

of Landsat imagery has grown continuously [24], mainly since 2008. The fact that the acquisition of remote sensing data—Landsat and other satellite images—coincided with the rapid urbanization taking place in Asian countries made several studies possible regarding this topic, i.e., [25]. Of course, Hanoi and Ho Chi Minh City (HCMC) are the two most investigated cities in Vietnam: many authors focused on their rapid growth and accompanying periurbanization. For example, [26] used demographic and remote sensing data to study urban transition in Vietnam and classify municipalities as rural, periurban, urban or urban core. The temporal series from 1992 to 2010 of a low-spatial resolution satellite (a weather sensor mounted on the Defense Meteorological satellite) shed light upon the constant growth of urban areas and rural villages, but mainly upon the big increase in periurban areas over the same period [27]. With regard to Hanoi, on the other hand, the studies made using remote sensing data are quite heterogeneous: for instance, they have used Landsat imagery to investigate the changes in the River Red section flowing though Hanoi [28]; to trace the increasing impact of urban sprawl on the urban heat islands in the metropolitan area [29], sometimes accompanied by a decrease in vegetation [30]; to map the chlorophyll-a concentration in hypertrophic waters like those of West Lake in Hanoi [31], etc. However, studies have usually turned their attention to the recent urban expansion of this city, also in comparison with other urban areas like Nagoya, Hartford and Shanghai [32]. Until the beginning of this century, the spatial growth of Hanoi was partially limited by natural barriers such as streams, water bodies and swamp areas [33,34]. Combining spatial metrics with remote sensing data (Landsat and ASTER images) [35] highlights how the expansion and outlying growth patterns prevailed between 1975 and 2001, while infill patterns were more important between 2001 and 2003. The rapid urbanization, which mostly occurred between 2001 and 2010, has also been highlighted by most studies, such as that of [36]. They investigate the urban growth of Hanoi at the provincial level, using multi-temporal image stacks of Landsat Thematic Mapper ™ and Enhanced Thematic Mapper (ETM+) images (1993–2010 period) and population data from 1999 and 2009 as datasets. Their classification shows how the great urban expansion of the capital occurred mainly between 10 and 25 km from the city center. The greatest change was in the periurban and rural communities, as previously classified by [26]. In these areas, there were significant demographic changes, such as an increase in population density in a 10 km radius around the city center. The dynamics of urban growth in Hanoi province have been closely examined by [37]: they used a rural–urban gradient analysis to compare the metrics and other indexes of several spatial patches (estimated on the basis of the same database mentioned previously, period 1993–2010) and their results support the diffusion–coalescence theory of urbanization for Hanoi city.

As such, this paper too aims to detail and offer an update on the changes in land use/land cover (LULC) which have taken place in Hanoi and its surroundings in the aftermath of Doi Moi (1989–2019). The main purpose is not only to highlight the magnitude of the recent impressive growth of Hanoi, but also to reflect upon the relevance of remote sensing for urban planners in order to understand the dynamics and spatial patterns of urban expansion. For these reasons, the first step was to define the study area: this was not a single administrative district (such as a region or province), but consisted of all the administrative districts within a 30 km radius of the city center. The area is characterized by similar geographic conditions: it is a rural plain, well connected to the central urban area, surrounded by modest altitude reliefs to the north and northwest, crossed by a main river (the Red River), but rich in secondary waterways. In order to study the effects of urban growth, three hypothetical rings were identified: the core (within a 10 km radius), the first ring (between 10 and 20 km from the centre) and, finally, the outer zone (over 20 km from the centre). For this, we collected and classified four Landsat images, approximately every ten years (1989, 2000, 2010 and 2019). Working on a topographic scale (i.e., [38]) in a Geographical Information System (GIS), it was possible to map—and understand—where the main changes in land cover have occurred in Hanoi during the last three decades. We then quantified the territorial development by decade and by ring to trace the progress of the periurbanization. Finally, using various different spatial indexes (the rate of urban expansion, some landscape metrics and the landscape expansion index), we examined the main

landscape dynamics currently in progress. The paper is structured as follows: the study area, dataset and methodology are described in Section 2 (Materials and Methods). Section 3 presents and discusses our results, focusing on the classification of remote sensed data and on the dynamics of urban growth. In conclusion, Section 4 contains some final remarks.

## 2. Materials and Methods

### 2.1. Study Area: Hanoi and Its Surrounding

Hanoi is located at longitude of 106° E and latitude of 21° N, in the heart of the Red River Delta (northern Vietnam). It is the country's most densely populated region with over 22 million inhabitants living in an area of about 21,000 km$^2$ (more than 8 million of whom live in urban areas) [5]. Officially founded in 1010 on a bend in the Red River, the Vietnamese capital has had a troubled history, which has affected its urban fabric: one thousand years of Chinese rule (1st century BC–10th century AD), the indigenous Vietnamese rule between the 11th and the 19th centuries, French colonial rule (1854–1954), the American war (1954–1975), the Soviet era (until the mid-1980s) and the current Doi Moi period have all contributed to shaping this city [39]. Since 2012, the Hanoi metropolitan area has included the city of Hanoi and nine other provinces (Bac Ninh, Bac Giang, Ha Nam, Hai Duong, Hoa Binh, Hung Yen, Phu Tho, Thai Nguyen and Vin Phuc). Hanoi Province is divided into 12 urban districts, 1 district-leveled town (or simply town) and 17 rural districts. As is well known, Hanoi has grown remarkably in recent decades, so much so that it is now the second largest city in Vietnam after Ho Chi Minh City, with more than 4 million people [5].

The area surrounding Hanoi is made up of plains close to the southeast coastline and is bordered to the north and west by hills and mountainous zones including some important national parks, such as Ba Vi and Tam Dao. Characterized by a humid, subtropical climate (hot summers and cold rainy winters), this is a region rich in water which includes not only the Red River, but many other waterways, ponds and lakes (such as the West Lake and the Hoan Kiem Lake). The town is surrounded by naturally irrigated agricultural land: rice (spring paddy and winter paddy) is the most important crop, followed by maize and vegetables [40]. Hanoi is one of the most ancient centers of Vietnam. Its four inner city districts (Hoan Kiem, Ba Dinh, Hai Ba Trung and Dong Da) are the heart of the town, also including the historical old quarter. This is the early core of Hanoi, located outside the Imperial Citadel of Thang Long.

However, the recent rural–urban and socio-demographic changes which took place mainly after the Doi Moi have turned this area into a very heterogeneous, complex territory. In order to study this region, we defined a specific circular zone around the center of Hanoi with a radius of about 30 km (Figure 1). Several criteria were involved in making this decision: it is a uniform geographical region; previous studies, e.g., [36], have highlighted how the urban growth has mainly occurred within this distance from the center; a significant amount (about 70%) of Hanoi's vegetable foodstuffs comes from local agricultural production in a radius of 30 km around the city [41]; moreover, new industrial and technological investments in the Hanoi metropolitan area are often made within a radius of 20 to 30 km of the city center [14].

For all these reasons, we defined our study area by selecting all the districts which fell within this zone (Figure 1), which covers about 3050 square kilometers, and lies between Hanoi, Bach Ninh and Hung Yen provinces. The area is located between the ranges of 105.5°–106.1° E and 21.4°–20.7° N and comprises 12 urban districts (Ba Dinh, Bac Tu Liem, Cau Giay, Dong Da, Ha Dong, Hai Ba Trung, Hoan Kiem, Hoang Mai, Nam Tu Lien, Long Bien, Tay Ho and Thanh Xuan), 3 towns (Bac Ninh, My Hao and Tu Son) and 20 rural districts (Chuong My, Dan Phuong, Dong Anh, Gia Lam, Hoai Duc, Khoai Chau, Me Linh, Phuc Tho, Quoc Oai, Soc Son, Thach That, Thanh Oai, Thanh Tri, Thuan Thanh, Thuong Tin, Tien Du, Van Giang, Van Lam, Yen My and Yen Phong) (Figure 2).

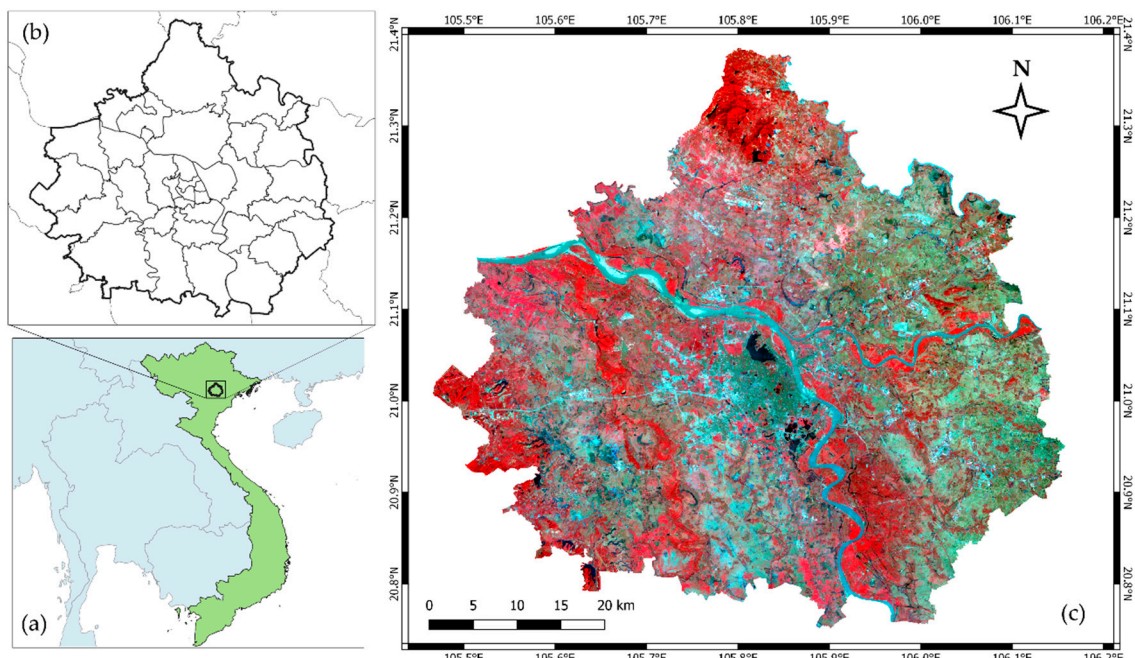

**Figure 1.** Location of the study area in Vietnam (**a**); Hanoi and its surrounding districts within a radius of 30 km (**b**); the 2010 Landsat 5 Thematic Mapper (TM) image false color composite (RGB: 432), clipped to the extent (coordinate system: WGS84) of the study area (**c**).

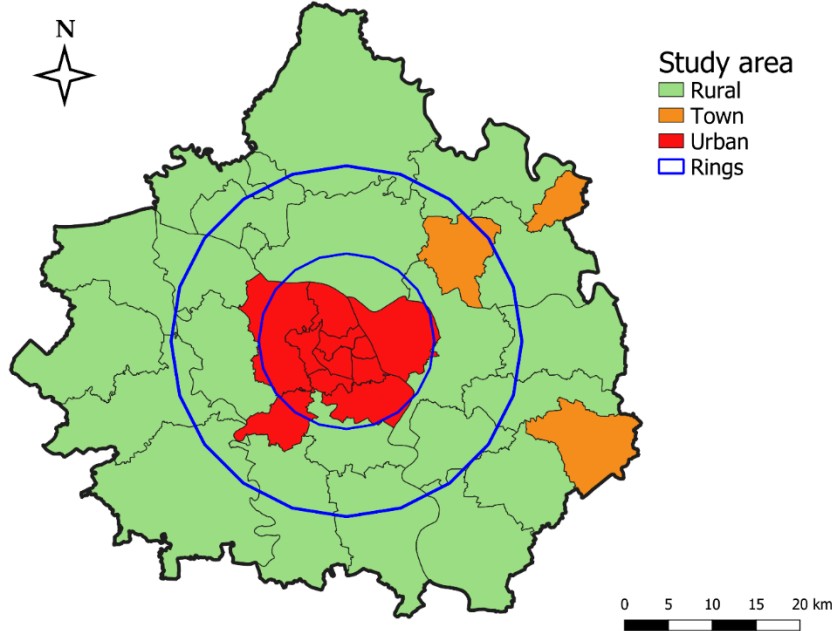

**Figure 2.** The classified districts of the study area (rural, town and urban), overlaid by two rings (10 and 20 km from the center, respectively), which identify the three concentric areas (core, first core and outside area).

## 2.2. Data Sources

The study involved classifying a miniseries of Landsat images of Hanoi acquired in the 1989–2019 period in order to examine its urban growth and the impact on the surrounding areas over these three decades. To achieve this goal, we used medium-resolution satellite images (to be precise, four pairs of Landsat images), one about every ten years. Two consecutive satellite sensor images had to be used to cover the entire study area (the Landsat World Reference System (WRS)—fort the

row is 045 for the northern image and 046 for the southern image of each pair). All the Landsat images were collected in the autumn season because they are usually the only ones available without cloud coverage (0–8%). Additionally, attention was paid to ensure that the study area was clear of clouds. All these images are "surface reflectance level-2 data products", delivered in GeoTIFF format (the standard format for the satellite images). The Coordinate Reference System (CRS) of all these products is WGS84/UTM 48N (EPSG code: 32648). These four pairs of Landsat scenes were supplied by the United States Geological Survey (USGS) through Earth Explorer (https://earthexplorer.usgs.gov/). The remote sensing database is detailed in Table 1.

**Table 1.** The satellite image database.

| Satellite | Sensor | Date of Acquisition | WRS -Path | WRS-Row | Cloud Coverage (%) |
|-----------|--------|---------------------|-----------|---------|--------------------|
| Landsat 5 | Thematic Mapper (TM) | 1989-11-30 | 127 | 045; 046 | 4; 8 |
| Landsat 7 | Enhanced Thematic Mapper (ETM+) | 2000-11-04 | 127 | 045; 046 | 0; 1 |
| Landsat 5 | Thematic Mapper (TM) | 2010-11-08 | 127 | 045; 046 | 1; 1 |
| Landsat 8 | Operational Land Imager sensor/Thermal Infrared Sensor (OLI/TIRS) | 2019-09-30 | 127 | 045; 046 | 0; 1 |

As cartographic maps, we also considered the polygon vectors of the district level of Vietnam, freely available from the Database of Global Administrative Areas (http://www.gadm.org/country), a high-resolution database of country administrative areas created in 2009 with the aim of providing vectors of "all countries, at all levels, at any time period".

### 2.3. Data Processing

The pre-processing procedures on the remote sensing data comprised: (a) the mosaicking of each single pair of images acquired in the four periods (1989, 2000, 2010 and 2019), and (b) the clipping of mosaicked scenes, using the polygon vectors of the study area.

We then applied a supervised classification (rule classifier: Maximum Likelihood Classification) in order to identify the following seven land use/land cover (LULC) classes: (a) water (rivers, streams, lakes, ponds, water reserves, etc.); (b) vegetated areas (forests or grassland partially covered with bushes and trees); (c) bare soil (sand along the main river or surfaces without vegetation); (d) cultivated areas (mainly rice fields or vegetable); (e) discontinuous built-up areas (rural village or dispersed urban fabric with widespread presence of green areas); (f) urban areas (very concentrated urban texture or continuous residential urban fabric); (g) industrial and commercial complexes.

The accuracy assessment procedure was assessed using 150 randomly stratified chosen ground control points (GCP) on each classified Landsat scene; to calculate the overall sample size, we chose at least of 20 GCP for a larger class [42]. Google Earth images were used as reference images to perform accuracy assessments of the 2000, 2010 and 2019 classified maps; we usually worked on a scale of detail of about 1:5000, using not only the newest images, but also the oldest images in order to understand the recent changes in landscape in the study area. For the oldest classified Landsat scene (1989), our reference images consisted of some old Vietnam topographic maps (scale 1:50,000, Series L7014), which had been made by the U.S. Army Map Service between the 1960s and early 1980s, freely available on the Perry-Castañeda Library Map Collection of the Texas Geodata Portal, University of Texas Libraries (http://legacy.lib.utexas.edu/maps/topo/vietnam/). In more detail, we georeferenced the following map sheets: Bac Ninh (6251-3, 1967), Ha Dong (6150-1, 1963), Ha Noi (6151-2, 1984), Luong Son (6150-4, 1984), Ke Sat (6250-4, 1966) and Son Tay (6151-3, 1966).

### 2.4. Urban Expansion Dynamics

Hanoi is still a fairly monocentric city, originally growing out from its historical center. To highlight the differences between the urban center and the surrounding rural areas, we adopted a GIS-based buffer analysis [43], involving circular buffer zones surrounding the city center. We took a central

point located in the historical old quarter of Hoan Kiem district (the Dong Kinh Nghia Thuc square) and we created three different concentric areas around it (Figure 2): (1) the core of the town, a circle of 10 km from the center including its historical center and the new urban districts; (2) the first ring, an area between 10 and 20 km from the center with all the more recent districts surrounding Hanoi city; and (3) the outside area, the zone more than 20 km from the center. We chose the 10 km buffer zone because the urban districts of the town were encompassed within this distance; furthermore, by using a 10 km radius, we could define the "near periurban" area (where the inner core meet the outer zone) and the "far periurban" zone well [18].

To quantify the scale and impact of urbanization, we made a visual comparison of the four classification maps. We then computed the changes in land cover classes occurring every ten years in the study area and in the three concentric areas above (core, first ring and outer ring). We also calculated the annual rate of urban expansion (*RUE*), some landscape metrics and the landscape expansion index (LEI) for spatial analysis of the urban dynamics.

### 2.4.1. The Rate of Urban Expansion (*RUE*)

The annual rate of urban expansion (*RUE*) [44] was calculated as given in Equation (1):

$$RUE = \frac{(BUA)i + n - (BUA)i}{n \times (BUA)i} \times 100 \tag{1}$$

where $(BUA)_{i+n}$ and $(BUA)_i$ are the building urban areas at time *i* + *n* and *i*, respectively, and n is the interval of the evaluating period. This is expressed as a percentage value per year. We used this index in order to evaluate the spatial distribution of urban expansion intensity during the time considered. Moreover, for this reason, the *RUE* was not only estimated for the whole study area, but also in the three rings (core, first ring and outer ring) for each of the decades considered.

### 2.4.2. The Landscape Metrics

Landscape metrics are commonly used to study spatial analysis. In this study, we considered the following four patch metrics: firstly, the number of patches (NP), namely their sum total in the landscape; secondly, edge length (EL), namely the sum of the lengths (m) of all the edge segments in the landscape; thirdly, mean patch area (MPA), expressed in square meters; and fourthly, the largest patch index (LPI), which is the percentage (%) of the landscape covered by the largest patch. We used these indexes because they are basic for spatial analysis and they are quite simple to understand. These spatial metrics are useful for the evaluation of urban dynamics: the hypothetical growth model of a town like Hanoi continuously alternates between processes of dispersion and coalescence, which means that the scale of analysis changes over time [36]. Dispersion leads to increasing fragmentation of the urban landscape, while the subsequent fusion of these growing scattered patches surrounding the city takes place in the phase of coalescence. The greater number of patches and longer edge length values could mean an increasing spread in the urban landscape during the years 1989–2019. The opposite is true for the mean patch area and largest patch index, whose coalescence increases.

The patch indexes were computed using the LecoS (Landscape Ecology Statistics) plugin of Quantum GIS (QGIS) [45], based on FRAGSTATS (acronym of "fragmentation statistics") functions [46]. As in previous cases, we computed the spatial metrics for the study area and for each ring in 1989, 2000, 2010 and 2019. All the landscape metrics used are shown in Table 2.

**Table 2.** Landscape metrics used in this study (from [37], mod.).

| Landscape Metrics | Description | Diffusion | Coalescence |
|---|---|---|---|
| Number of patches (NP) | Total number of patches in landscape | Increasing | Decreasing |
| Edge length (EL) | Sum of the lengths (m) of all edge segments in landscape | Increasing | Decreasing |
| Mean patch area (MPA) | Average area of patches (in sq. m) | Decreasing | Increasing |
| Largest patch index (LPI) | Proportion (%) of landscape covered by largest patch | Decreasing | Increasing |

### 2.4.3. The Landscape Expansion Index (*LEI*)

In 2010, [47] proposed a new landscape index for quantifying urban expansion using multi-temporal remotely sensed data in order to obtain information about the spatio-temporal dynamic changes in landscape patterns. They consider three main types of spatial pattern (Figure 3), namely infilling, edge expansion and outlying. Infilling occurs when the built up area increases in the urbanized open space between (or within) the old; edge expansion refers to a newly grown patch spreading unidirectionally in more or less parallel strips from one of the edges of the previous development; outlying (or spontaneous) growth concerns an isolated newly grown patch. We evaluated this index for the dense urban fabric class. The landscape expansion index (*LEI*) [47] was calculated as given in Equation (2):

$$LEI = \frac{Lc}{P} \times 100 \tag{2}$$

where *Lc* is the length of the common boundary of a newly grown patch area with an old building area and P is the perimeter of this new patch area. We applied a buffer zone of 200m. According to this definition, values of *LEI* > 50 mean infilling growth, while 0 < *LEI* <=50 refers to the edge expansion growth type and *LEI* = 0 refers to spontaneous growth. As for the previous indexes, for each decade considered, we estimated the *LEI* (number of patches and area) not only for the whole study area, but also in the three rings (core, first ring and outer ring).

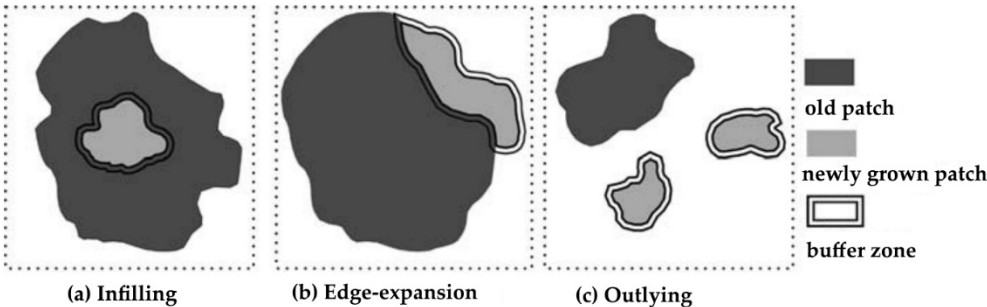

**Figure 3.** The three types of landscape expansion: (**a**) infilling; (**b**) edge expansion; (**c**) outlying or spontaneous (source: [47]).

## 3. Results and Discussion

### 3.1. Accuracy Assessment

The LULC maps (1989, 2000, 2010 and 2019) obtained by the classification of the Landsat images were validated from confusion matrices where the overall accuracy and kappa coefficient (k) were computed (Table 3). Based on [48,49], overall accuracy values and k values indicated a strong agreement between the ground truth and the classified classes. Usually, the producer and user accuracy scores of the identified land cover types (water, cultivated areas, etc.) were over 80%, with the occasional exception of those cover types that are less common in the study area such as bare soil or vegetated areas, where it was possible to assess only very few GCPs. With regard to the different LULCs, we usually achieved good results for all classes (average user and producer accuracy >80%), with the exception of industrial and commercial complexes in the classification of older satellite images (1989 and 2000). The best results are those related to the classification of cultivated areas (average user accuracy: 93%) and discontinuous built areas (average producer accuracy: 94%).

**Table 3.** Overall accuracy and K values of the classified Landsat images (1989, 2000, 2010 and 2019).

| Year | Overall Accuracy | K Value |
|------|------------------|---------|
| 1989 | 86% | 0.813 |
| 2000 | 84% | 0.779 |
| 2010 | 85% | 0.800 |
| 2019 | 88% | 0.846 |

### 3.2. The Classified Images: Some Significant Statistics

The miniseries of LULC maps (Figures 6–9) show the main changes occurring in Hanoi and its surrounding area over the last three decades. Figure 4 provides the percentage of each land cover class in the different years (1989, 2000, 2010 and 2019), also reported in detail in Table 4. Table 5 gives the information regarding use of land cover as percentage changes during the different time intervals (1989–2000; 2000–2010; 2010–2019; 1989–2019). Figure 5 shows the percentages of each class by ring (core, first ring and outside) between 1989 and 2019, full details of which are also provided in Table 6.

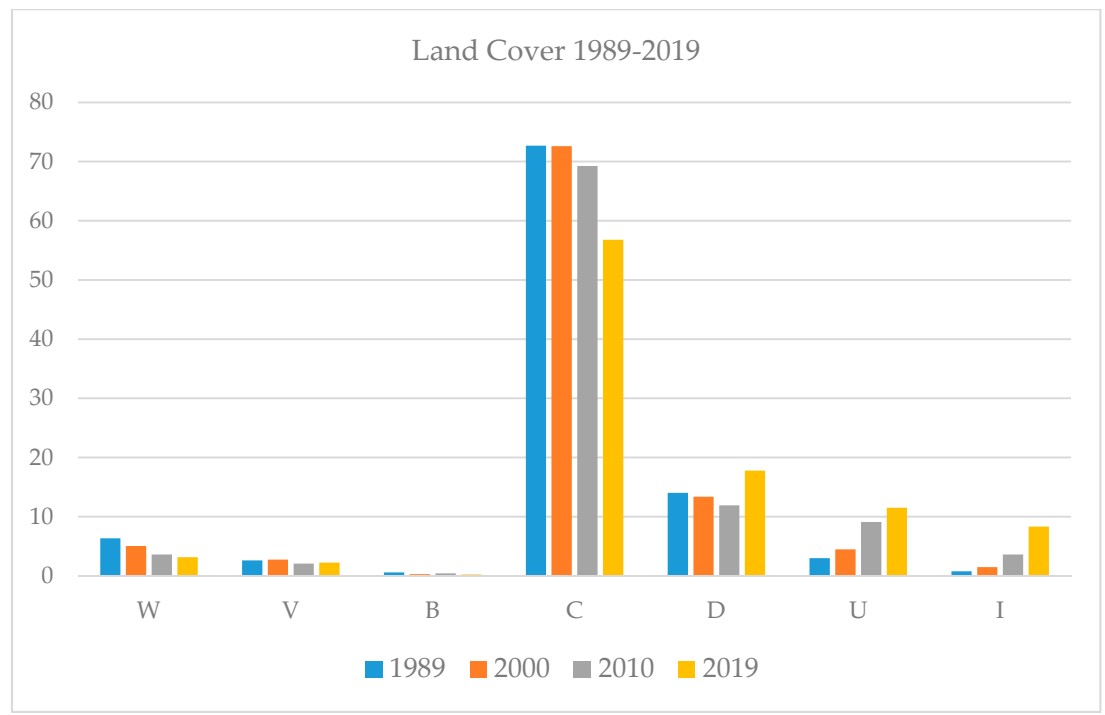

**Figure 4.** Land use/land cover (LULC) classes (in percentages) in the study area between 1989 and 2019 W: water; V: vegetated areas; B: bare soil; C: cultivated areas; D: discontinuous built-up areas; U: urban areas; I: industries and commercial complexes).

**Table 4.** LULC classes (square kilometers and percentages) in the study area between 1989 and 2019.

| Land Cover | 1989 | | 2000 | | 2010 | | 2019 | |
|------------|------|---|------|---|------|---|------|---|
| | km² | % | km² | % | km² | % | km² | % |
| W | 193.9 | 6.4 | 153.7 | 5 | 110.2 | 3.6 | 95.9 | 3.1 |
| V | 79.7 | 2.6 | 83.3 | 2.7 | 63.3 | 2.1 | 68.2 | 2.2 |
| B | 17.6 | 0.6 | 8.6 | 0.3 | 13 | 0.4 | 7.1 | 0.2 |
| C | 2217.1 | 72.7 | 2214.9 | 72.6 | 2112.5 | 69.3 | 1731.9 | 56.8 |
| D | 427.4 | 14 | 407.9 | 13.4 | 363.4 | 11.9 | 542.6 | 17.8 |
| U | 91.1 | 3 | 136.7 | 4.5 | 277.4 | 9.1 | 351.1 | 11.5 |
| I | 23.2 | 0.8 | 44.9 | 1.5 | 110.3 | 3.6 | 253.3 | 8.3 |
| Total | 3050.1 | 100 | 3050.1 | 100 | 3050.1 | 100 | 3050.1 | 100 |

**Table 5.** LULC changes (as percentages) in the study area between 1989 and 2019.

| Land Cover | 2000–1989 | 2010–2000 | 2019–2010 | 2019–1989 |
|:---:|:---:|:---:|:---:|:---:|
| W | −1.4 | −1.4 | −0.5 | −3.3 |
| V | 0.1 | −0.6 | 0.1 | −0.4 |
| B | −0.3 | 0.1 | −0.2 | −0.4 |
| C | −0.1 | −3.3 | −12.5 | −15.9 |
| D | −0.6 | −1.5 | 5.9 | 3.8 |
| U | 1.5 | 4.6 | 2.4 | 8.5 |
| I | 0.7 | 2.1 | 4.7 | 7.5 |

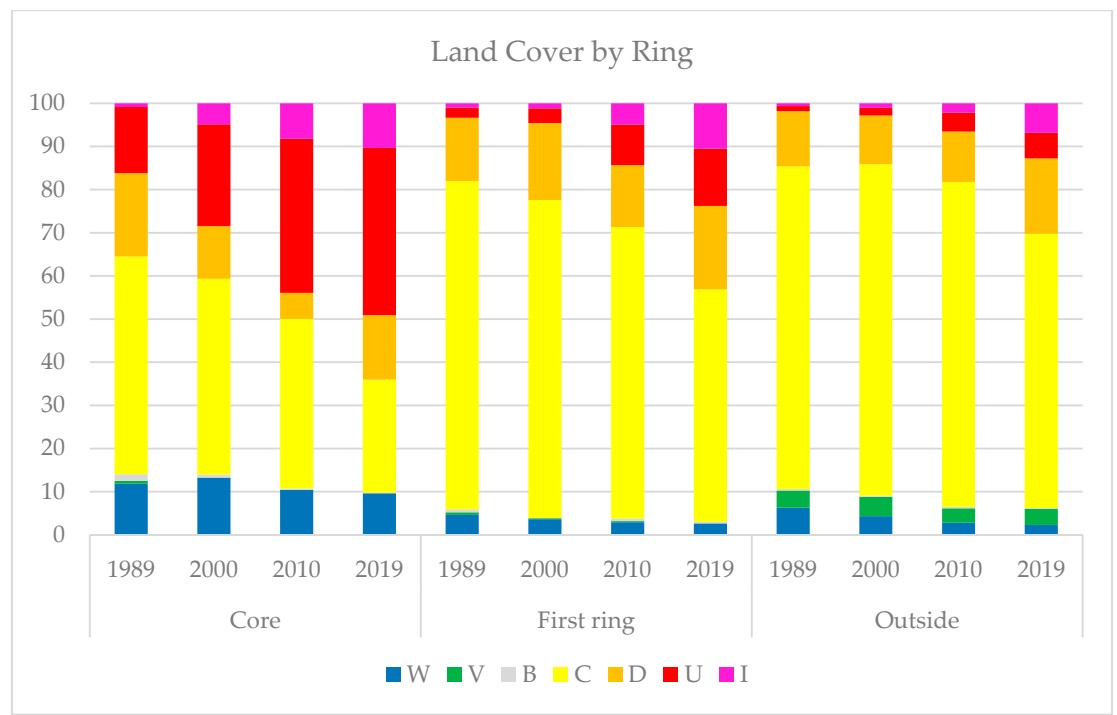

**Figure 5.** LULC classes (as percentages) in the three rings between 1989 and 2019 (WW: water; B: bare soil; V: vegetated areas; C: cultivated areas; D: discontinuous built-up areas; U: urban areas; I: industries and commercial complexes).

**Table 6.** LULC (as percentages) by rings between 1989 and 2019.

|  | Year | W | V | B | C | D | U | I |
|:---:|:---:|:---:|:---:|:---:|:---:|:---:|:---:|:---:|
| Core | 1989 | 11.9 | 0.7 | 1.5 | 50.5 | 19.3 | 15.3 | 0.9 |
|  | 2000 | 13.3 | 0 | 0.7 | 45.4 | 12.2 | 23.6 | 4.9 |
|  | 2010 | 10.4 | 0 | 0.4 | 39.3 | 6.0 | 35.7 | 8.2 |
|  | 2019 | 9.6 | 0 | 0.1 | 26.2 | 14.9 | 38.8 | 10.3 |
| First ring | 1989 | 4.6 | 0.6 | 0.7 | 76.1 | 14.7 | 2.3 | 1.0 |
|  | 2000 | 3.6 | 0.2 | 0.1 | 73.6 | 17.9 | 3.4 | 1.2 |
|  | 2010 | 2.9 | 0.3 | 0.6 | 67.5 | 14.3 | 9.4 | 4.9 |
|  | 2019 | 2.6 | 0 | 0.4 | 53.8 | 19.4 | 13.3 | 10.5 |
| Outside | 1989 | 6.3 | 4.0 | 0.4 | 74.7 | 12.8 | 1.2 | 0.6 |
|  | 2000 | 4.4 | 4.5 | 0.3 | 76.7 | 11.3 | 1.8 | 1.1 |
|  | 2010 | 2.8 | 3.4 | 0.3 | 75.3 | 11.7 | 4.4 | 2.2 |
|  | 2019 | 2.3 | 3.7 | 0.2 | 63.5 | 17.5 | 5.9 | 6.8 |

Overall, although the landscape remained predominantly rural in nature, over the period 1989–2019, the size of the cultivated areas decreased by almost 500 km$^2$ (meaning about 16% less than 1989). This partly explains the significant loss of almost 100 km$^2$ of water, whose presence in this area

is closely connected to the rice crop. Over the same period, anthropized landscapes such as urban districts, discontinuous built-up areas and industrial or commercial complexes increased by more than 600 km$^2$ (about 20% more compared to 1989). More specifically, urban areas increased by over 250 km$^2$, while warehouses and industries increased tenfold in just a few years (from 23.5 to 253 km$^2$).

These LULC changes took place following differing gradual patterns from the core to the periphery. The inner core of the study area was most greatly affected by the process of urbanization and industrialization; there, we see a concomitant striking reduction in agricultural areas, and also in discontinuous built-up areas. Although the values are lower, similar territorial dynamics seem to have occurred in the first and the outside ring, but these are still predominantly rural. Vegetated areas (i.e., forests) had a significant presence only in the external ring, while bare soil was quite a marginal class.

We shall now provide a brief description of land cover changes in the study area between 1989 and 2019, on the basis of our observations of the LULC maps and the statistics we came up with.

### 3.2.1. The LULC in 1989

The densification of the urban area of Hanoi is already evident in the 1989 LULC map, as has also been pointed out by [8,32], with the area changing from a small, fragmented town in 1975, developing mainly to the south of the old quarters, to a compact city located on the right bank of the Red River by 1989 (Figure 6). The infill process is almost complete in its historical districts (i.e., Bah Dinh, Dong Da, Hai Ba Trung and Hoan Kiem). During this period, the two most significant elements are the densification of the urban area and its first extension along the most important road axes. The industrial zones (0.8%) are usually located in the outlying areas of the old city along the main road infrastructure [50].

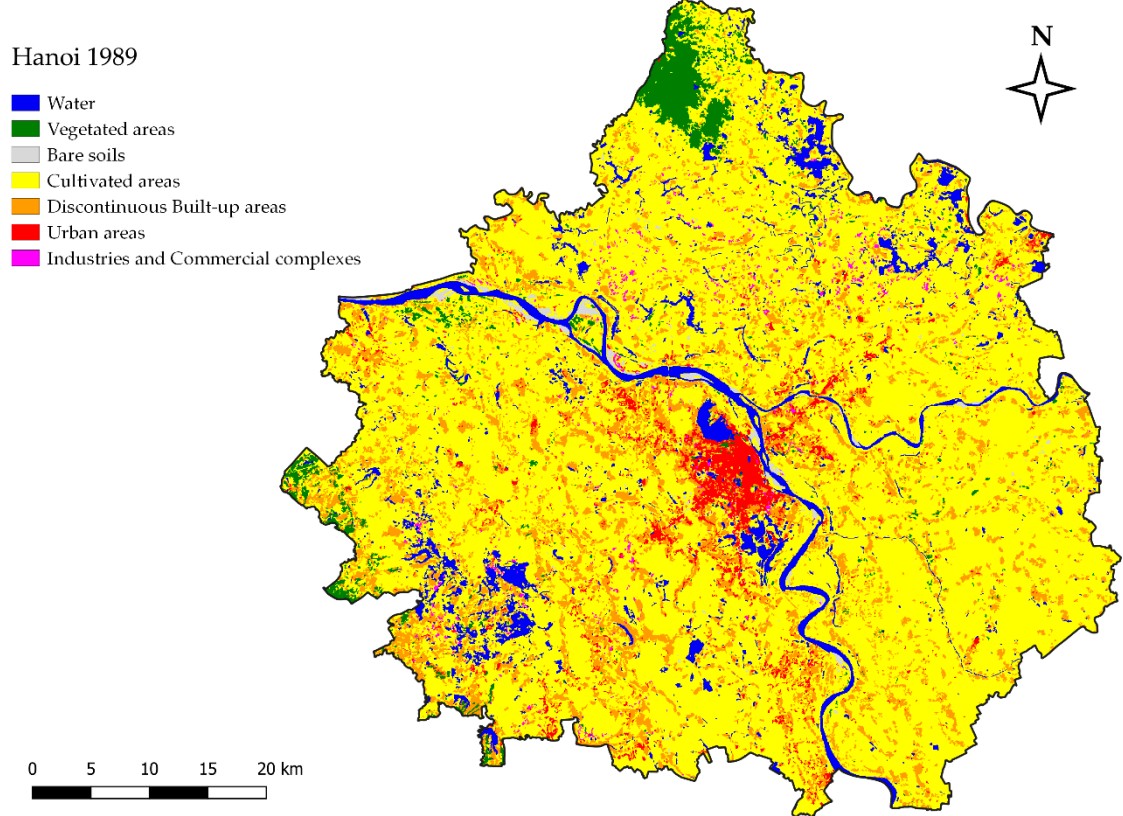

**Figure 6.** The 1989 LULC map of the study area.

However, in 1989, the surrounding area was still a predominantly rural region: more than 70% of the study area was cultivated land, and rice paddies in the satellite image often appeared covered by water (over 6% of study area). The cultivated fields covered over 74% of the first and the outside rings, but were also prevalent within the core (more than 50%). The discontinuous built-up area was already quite significant (14% of study area), mainly within the two inside rings (representing 19.3% in the core and 14.7% in the first ring), but scattered fairly evenly throughout the entire study area. The forest (classified as vegetated areas) was marginal and located mainly to the north, covering part of the current Tam Dao National Park. Bare soils (about 0.6% of study area) were to be found within or close to the Red River in the core, while the presence of sheds and industrial classes was negligible.

### 3.2.2. The LULC in 2000

The first steps towards a more organized territory can be observed in the LULC map of 2000 (Figure 7): while the surrounding area seems to never change, Hanoi becomes even more compact. In this decade, the spatial structure of the city underwent a remarkable transformation, mainly due to the increasing value of the housing in the inner core of the city [51]. Rapid expansion of the periphery occurred regularly in the northeastern area (close to Lake Hò Tây [34]) and in the southwestern area of the city, but it also began during those same years on the east bank of the Red River, as has been highlighted by [35]. Moreover, the city began to grow radially during this period along the new road infrastructure in several directions such as to Chuć Son in the southwest, to Thuong Thin in the south or, again, to Bach Ninh in the northeast. The proportion of discontinuous built-up areas in the territory stays almost the same (13.4%) as in 1989. It decreases in the core, however (from 19.3% to 12.2%), often becoming dense urban fabric, while increasing in the first ring (from 14.7% to 17.9%).

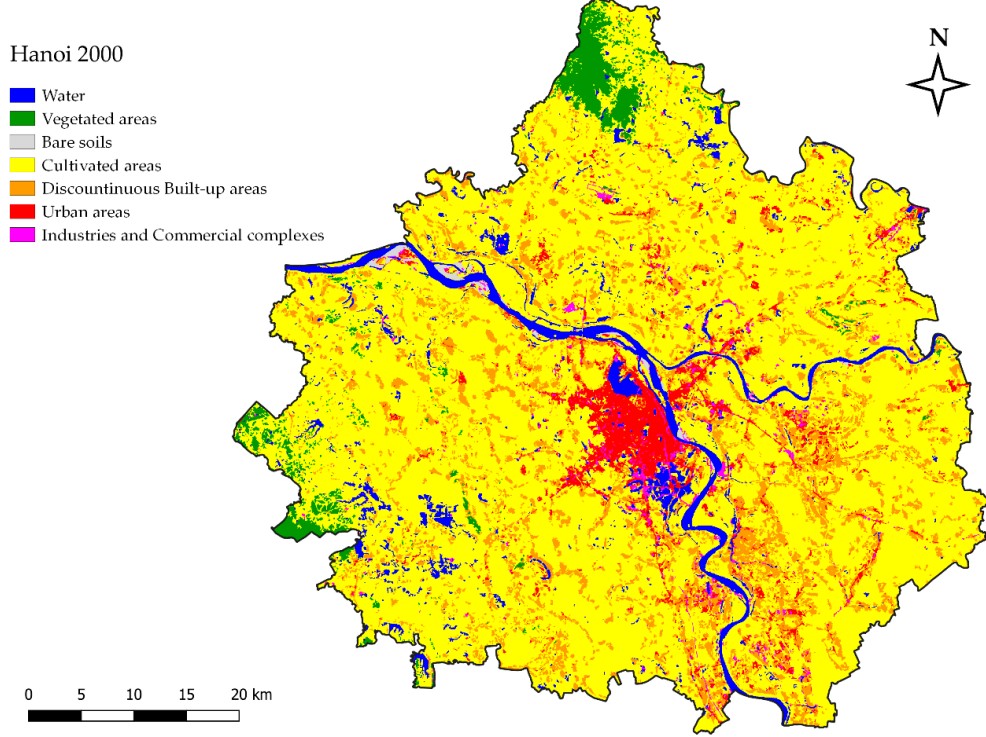

**Figure 7.** The 2000 LULC map of the study area.

Nevertheless, the region remains mainly rural: agricultural land still accounts for over 70% of the entire study area. However, the core proves to be more dynamic than the other rings between 1989 and 2000. While the agricultural areas account for almost 75% of land use in the two external rings, in the area closest to the capital there is a significant increase in industries and commercial complexes (from 0.9% to 4.9%) and in the urban dense fabric class (from 15.3% to 23.6%). As clearly shown by [50],

the new industrial zones are usually located in isolated rural areas. More specifically [39] (p. 170), they are located in three outer districts (Gia Lam, Sóc Son and Dóng Anh), in pursuit of the goals of the Strategic Planning approach, which "aim to develop Hanoi as a centre of wholesale market services, export and as a service centre". Land cover changes regarding the other classes (water, forest and bare soils) are minor.

### 3.2.3. The LULC in 2010

Here, the situation has changed greatly in just a few years: the 2010 LULC map (Figure 8) shows how the rural context surrounding Hanoi has been transformed into a more complex territory. The city has become less compact: suburban housing, industries and infrastructure in the outlying areas make this periphery a typical rural–urban fringe area where the border between the city and its surroundings is less defined. The town continues to expand along the western edge with new and improved infrastructure, and with the appearance of new residential, leisure and industrial complexes [10]. Its growth also proceeds radially, in different directions from the year 2000, such as to Ha Dong in the southwest. In only ten years, the urban areas have doubled in size (from 4.5% to 9.1%), due largely to the city expanding, but also to an accompanying increase in the urban area class in the adjacent areas (first ring and outside). The growth in dense urban fabric mainly relates to the core (from 23.6% to 35.7%) and the first ring (from 3.4% to 9.4%), where, at the same time, there is a decrease in the discontinuous built-up areas (in the core from 12.2% to 6%; in the first ring from 17.9% to 14.3%). The expansion of the city reshaped and often wiped out the pre-existing villages: this mixing of new urban villages with the former rural villages results in a new process of urban formation called "villages in the city" in the local language [52]. Thus, the resettlement of new urban zones is often achieved through new high-income settlements being "inserted" adjacent to the former rural villages [53] (p. 120), in a sort of "hybrid transitional system", the result of an alliance between local government and the private sector [54] (p. 79).

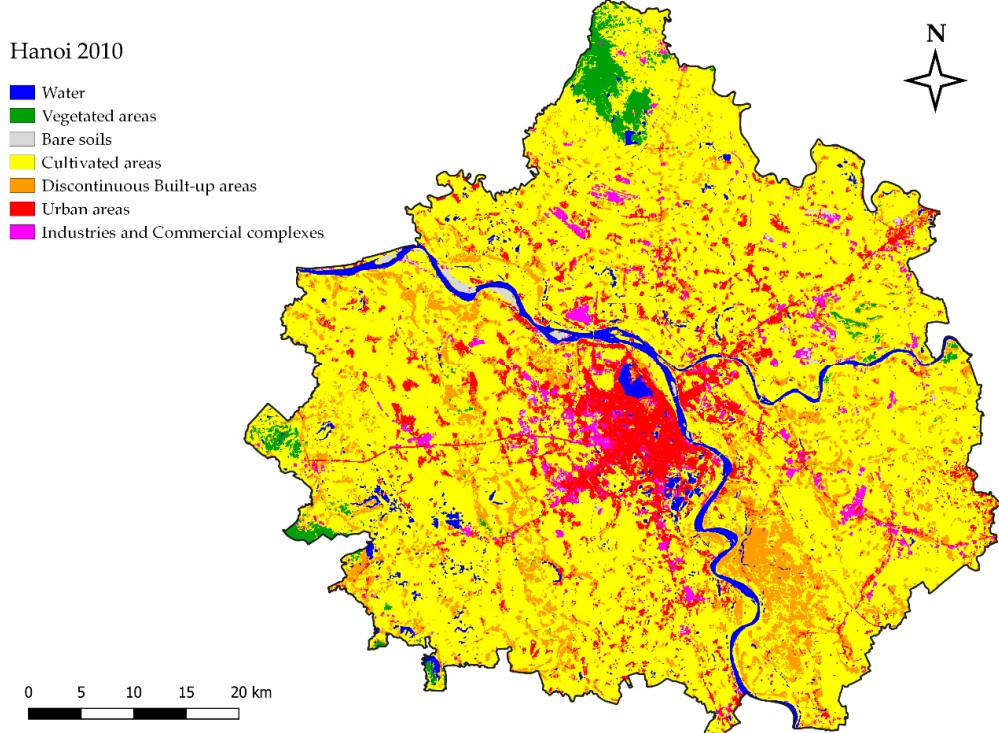

**Figure 8.** The 2010 LULC map of the study area.

Alongside urbanization, the years from 2000 to 2010 also saw an impressive growth in industrial and commercial activities, with a doubling of their land cover areas (from 1.5% to 3.6%). This usually

followed an urban–rural pattern: the growth in business premises and warehouses (i.e., factories, shopping malls, etc.) mostly occurred near the city (from 4.9% to 8.2%) and along the main roads in the first ring (from 1.2% to 4.9%), while in the outside area this increase was less significant (from 1.1 to 2.2%). Meanwhile, agricultural lands declined from 72.6% in 2000 to 69.3% in 2010, mainly in the core (less than 40%) and in the first ring (less than 68%). In the two external rings, the new infrastructure and the generalized growth in built-up areas (the sum of urban areas and discontinuous built-up areas) made the ancient rural landscape less uniform and more fragmented.

### 3.2.4. The LULC in 2019

The LULC map of 2019 (Figure 9) shows the extent to which the urban growth of Hanoi has changed the surrounding area in a very complex region. Overall, the study area is no longer a solely rural landscape: the remaining—but still prevailing—cultivated areas (almost 57%) are by now very fragmented and usually located at quite a distance from the city. The core of the study area is now predominantly occupied by urban areas (38.8%), discontinuous built-up areas (14.9%) and industrial and commercial complexes (10.3%). In the meantime, the expansion and the radial growth of the city has continued along the main roads, crossing the geographical boundary of the Red River, but also the administrative boundaries of the ancient urban districts. Thus, on the eastern bank of the Red River, the district of Long Bien became urban as early as 2003; the neighboring district of Gia Lâm, though still officially a rural district, is actually a suburb of eastern Hanoi with more than 200,000 inhabitants. To the east, the periurban area located between ring roads 3 and 4, within the districts of North Tù Liêm, South Tù Liêm and Ha Dong, is now almost completely urbanized with approximately 1.5 million inhabitants [55]. In actual fact, the entire study area is involved in the current urbanization: the built-up areas (meaning urban and discontinuous built-up areas combined) have seen remarkable growth both in the first ring (from 23.7% in 2010 to 32.7% in 2019), and in the outer ring (from 16.1% in 2010 to 23.4% in 2019).

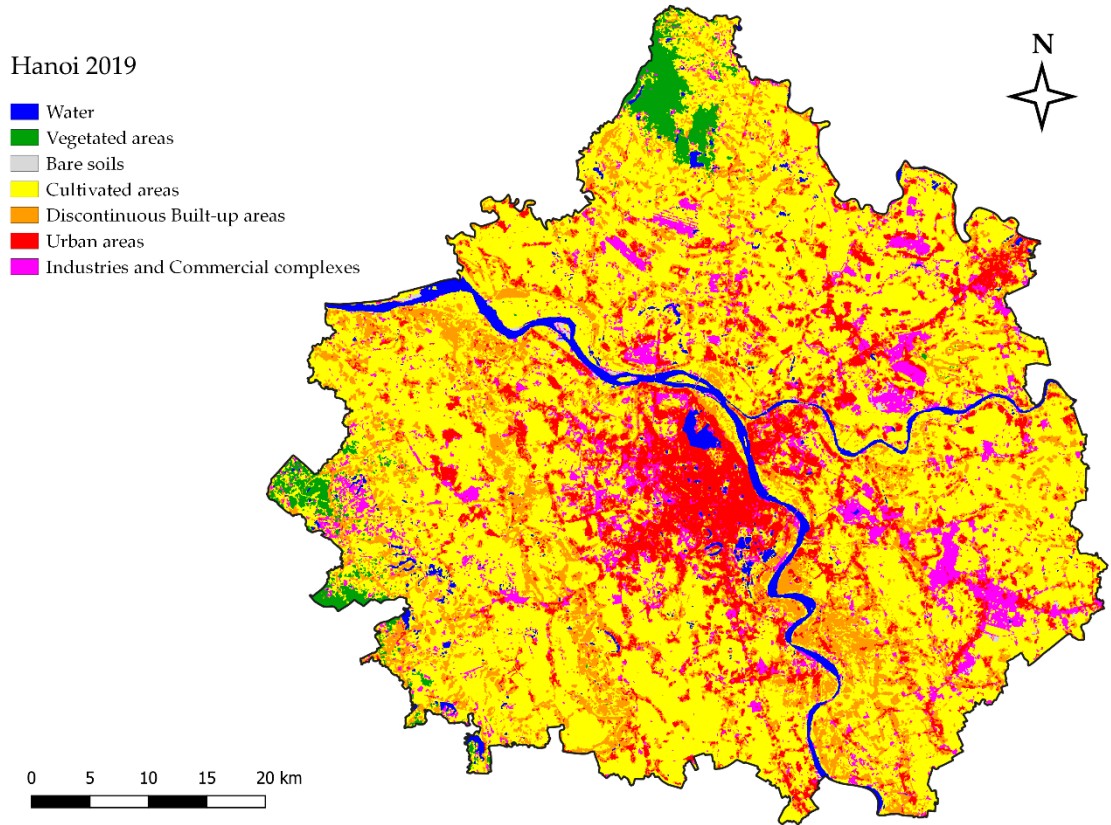

**Figure 9.** The 2019 land cover map of the study area.

Rapid industrialization is proceeding: over the last decade, the mix of industrial and commercial areas has taken over increasingly large areas both near Hanoi city and its surroundings, occupying more than 10% of the first ring and almost 7% of the outer ring. As highlighted by [56] (p. 204), "a significant area of rice fields, especially along major national highways have been converted to large factories and industrial parks". This is the case of the Pho Noi industrial park to the southeast of Hanoi, located along national highway n. 5, 25 km away from the town, which sprang up in just a few years, as can be seen from the satellite images of Google Earth (Figure 10).

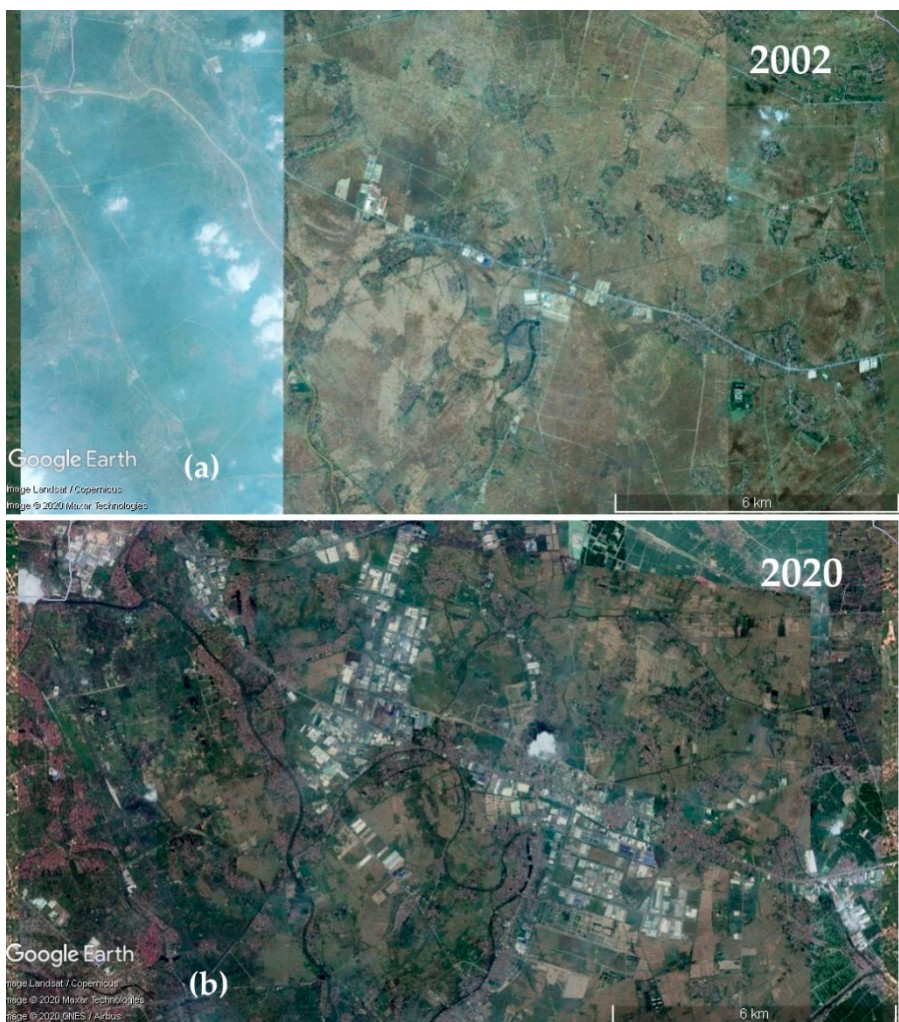

**Figure 10.** The Pho Noi area: (**a**) the rural region in 2012; (**b**) the industrial park in 2019 (source: Google Earth).

### 3.3. Delineating the Dynamics of Urban Expansion

Urban expansion from 1989 to 2019 occurred throughout the study areas: in less than thirty years, the area of land covered by urban areas almost quadrupled, from 91.1 to 350 km$^2$ (Table 7), sometimes just increasing the density of pre-existing discontinuous built-up areas. Between 1989 and 2019, the extent of dense urban fabric increased by 8.5% (by 12.3% if spread urban fabric is also considered), while agricultural land decreased by almost 16% (Table 5).

The annual RUE for the three ten-year periods considered (1989–2000, 2000–2010 and 2010–2019) showed an increase in the extent of urban areas of 4.5% (4.1 km$^2$/year), 10.3% (14 km$^2$/year) and 2.9% (8.2 km$^2$/year) respectively; with an average annual increase of 9.5% (8.7 km$^2$/year) for the period from 1989 to 2019 (Table 7).

**Table 7.** Rate of urban expansion (RUE) in the study area between 1989 and 2019.

| | Urban Area | Rate of Urban Expansion (% Year$^{-1}$) | | | |
|---|---|---|---|---|---|
| Year | km$^2$ | 1989–2000 | 2000–2010 | 2010–2019 | 1989–2019 |
| 1989 | 91.1 | 4.5 | | | |
| 2000 | 136.7 | | 10.3 | | |
| 2010 | 277.4 | | | 2.9 | |
| 2019 | 351.1 | | | | 9.5 |
| Change (km$^2$ year$^{-1}$) | | 4.1 | 14.1 | 8.2 | 8.7 |

The RUE's values by ring (Figure 11) show differing rates in the urban growth of the three identified zones (core, first ring and outside). RUE values are usually low in the core, with an average value between 1989 and 2019 of about 5% (2.4 km$^2$/year), while levels of RUE are higher in the first ring (average value: 15.6%, namely 3.4 km$^2$/year$^{-1}$) with a peak value of 17.8% (5.6 km$^2$/year$^{-1}$) in the period 2000–2010. The outer ring, too, showed high RUE values: there, the average percentage increase over the last three decades was 12.8% (2.8 km$^2$/year). To sum up, over the last three decades, urban growth has mainly occurred in the periurban area of Hanoi City, as a result of the conversion of agricultural lands and pre-existing rural villages.

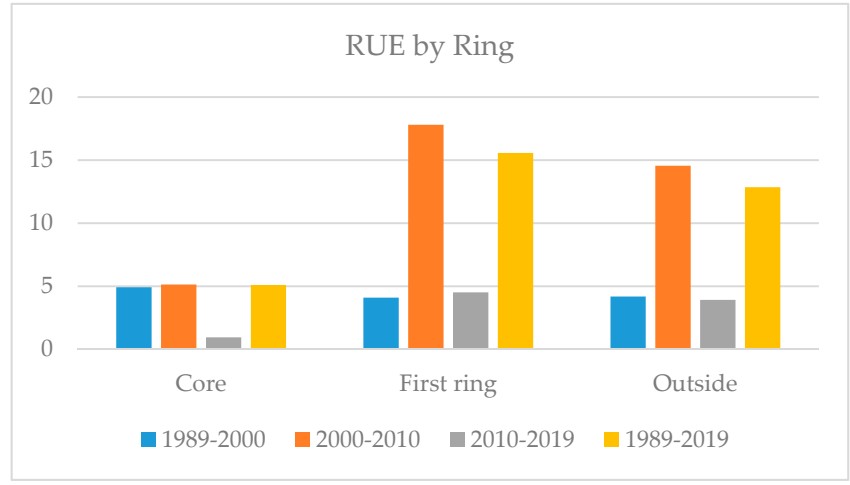

**Figure 11.** RUE values by ring between 1989 and 2019.

Figure 12 shows the landscape metrics (NP, EL, MPA and LPI) values for the urban class in the different years (1989; 2000; 2010; 2019), considering the entire study area as well the single rings. Nearly all these metrics show an increase, though they may increase in different ways. In the three decades of the study period, the NP of the study area rises from 1724 to 2578, but mainly in the two outer rings (from 629 to 995 in the first ring; from 708 to 1435 in the outside), while, in the core, there is a slight decrease (from 379 to 244) (Figure 12a). In line with this, the EL shows a significant increase if we consider the entire study area (from 2178 to 5350 km), with almost the same values for the two outer rings (from 670 to 2120 km in the first ring; from 740 to 2200 km for the outside), while the values of the core are almost unchanged (from 780 to 1080 km) (Figure 12b). At the same time, there is a noticeable increase in the MPA and LPI in the core (from 12 to almost 50 ha and from 9% to 26%, respectively), while, in the other two rings, there is only a slight increase (in MPA) or no change (LPI), from just over zero to 3%. In short, in the 1989–2019 period, there is usually steady growth in NP and EL in the study area, especially in the two outer rings, where the diffusion process is well under way. In the meantime, the urban patches of the core become increasingly large, occupying significant portions of the town and bringing about a process of coalescence between the urban areas of the center and the town's periphery.

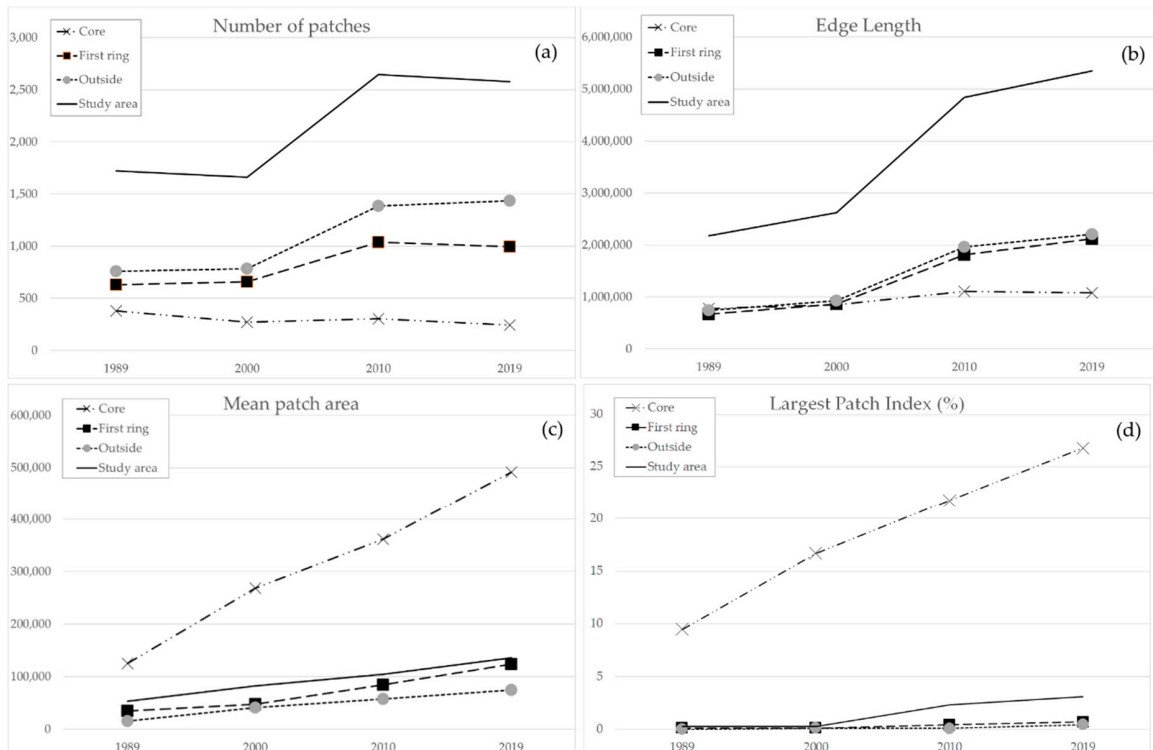

**Figure 12.** Landscape metric values in the entire study area and by ring in the 1989–2019 period: (**a**) number of patches (NP); (**b**) edge length (EL); (**c**) mean patch area (MPA); (**d**) largest patch index (LPI).

In order to better analyze the dynamics of urban expansion, it is useful to identify and quantify the different types of urban area. To do this, we classified the urban growth taking place in the city of Hanoi and its surroundings during the 1989–2000, 2000–2010 and 2010–2019 periods into three types: infill, edge expansion and spontaneous (Table 8). Table 7 shows that the new urban areas of about 45 km$^2$, which appeared during the 1989–2000 period, were composed of 7% of infill and almost 60% of edge expansion, while 33% were spontaneous (equivalent to 9.5%, 58.4% and 32.1% of new patches, respectively). During the 2000–2010 period, there was even more edge expansion (by about 63%, equivalent to 60% of the new patches) from the new 140 km$^2$ of urban areas, while new infill and spontaneous growth accounted for 5% and 32.3%, respectively. The upward trend in edge expansion continued in the 2010–2019 period (representing almost 74% of the new 74 km$^2$ of built-up area, equivalent to 68% of new patches), whereas the increase in infill (10.7%) and spontaneous (15.7%) was lower. The results show that edge expansion is the predominant type of urban growth in the study area, whereas infill is gradually becoming more important and spontaneous expansion is decreasing.

**Table 8.** Percentage values of number of patches and areas by type of building expansion (spontaneous, edge expansion or infill) in the three different periods (1989–2000, 2000–2010 and 2010–2019).

| Patches | Type | 1989–2000 | 2000–2010 | 2010–2019 |
|---|---|---|---|---|
| Number (%) | Infill | 9.5 | 11.3 | 18.5 |
| | Edge expansion | 58.4 | 60 | 67.9 |
| | Spontaneous | 32.1 | 28.7 | 13.6 |
| Area (%) | Infill | 6.9 | 4.8 | 10.7 |
| | Edge expansion | 59.4 | 62.9 | 73.6 |
| | Spontaneous | 33.7 | 32.3 | 15.7 |

For a more in-depth analysis, Figure 13 considers the percentage values of the different types of urban growth by ring and by type of building expansion. Moreover, in this case, edge expansion

was the most significant type, but the trend followed a different pattern depending on the ring; while it decreased in the core, the opposite was true for the two outer rings. The infill process became increasingly important in the core, while spontaneous expansion almost disappeared. By way of contrast, the first ring and the outside saw a slow decrease in spontaneous building, though this was still significant, especially on the outside; expansion by infill, on the other hand, is presently much less important in both of the outer rings.

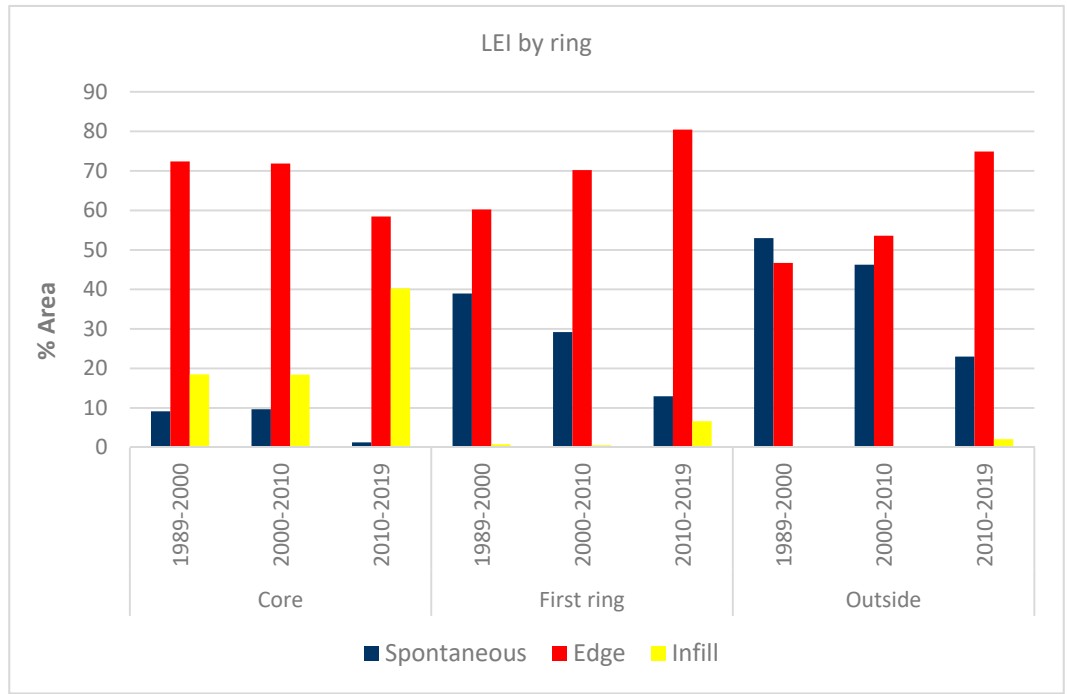

**Figure 13.** Percent values of area by ring and by type of building expansion (spontaneous, edge expansion or infill) in the three different periods (1989–2000, 2000–2010 and 2010–2019).

In accordance with [37], the LEI shows the dynamics of urban expansion in a similar way to that highlighted by landscape metrics. In fact, it is the infilling process that is increasingly important in the core, giving rise to coalescence in urban development between Hanoi's town and its closest suburbs. In parallel, in the outer rings, there is a decrease in the rate of expansion of new spontaneous built-up areas, whereas the edge expansion type is growing, meaning that a process of diffusion is under way in the periurban area.

## 4. Conclusions

Remote sensing represents a great opportunity to monitor contemporary urbanization. In particular, the free Landsat imagery acquired since the 1980s makes it possible to investigate urban expansion in those regions currently experiencing the fastest rates of urbanization, namely East and Southeast Asia. In this study, multi-temporal Landsat images for the years 1989, 2000, 2010 and 2019 were used to produce LULC maps of Hanoi on a topographic scale, to understand the dynamics and spatial patterns of its urban growth. We considered all the districts within a radius of 30 km from the center. Over the last three decades (1989–2019), the urban fabric in the study area increased around fourfold (from about 90 km$^2$ to 350 km$^2$), with higher rates in the closest periurban suburbs than in the core. In the downtown districts, on the other hand, edge expansion and infill growth of urban patches were the most prevalent types of expansion the last thirty years; as well as an increase in LPI and MPA landscape metrics, these spatial patterns highlight the ongoing process of fusion between urban patches, mainly in the core of our study area. While the 1990s saw a gradual densification of the pre-existing urban area, the town has rapidly expanded outwards since the early 2000s, often spreading

out radially along the axes of its main access roads, and ending up encompassing the neighboring rural villages. Confined for hundreds of years in a limited space on the right bank of the Red River, just like other newly globalized metropolitan areas, Hanoi is now a fringed city. It has spread with astonishing speed into the surrounding countryside, even bridging the geographical boundary of the river. In the current globalization era, in only few decades "city and countryside are redefined by the industrial logic of capitalist production and accumulation, losing much of their former substantive original traits" [57] (p. 336).

At the same time, the region around Hanoi is no longer rural: in just thirty years, urbanization has converted this territory into an industrial and commercial region. New social and economic policies after Doi Moi in 1986 have given rise to rapid industrialization in the study area, with a huge loss of farmland on the fringes of Hanoi. In the period 1989–2019, we estimated a loss of almost 500 km$^2$ of agricultural landscape out of a total area of 3050 km$^2$ (meaning 16% less than 1989); moreover, the rural landscape is now less uniform and more fragmented than thirty years ago. In the meantime, industrial and commercial areas have grown steadily: while at the end of the last century, industrial parks were usually placed near the town, they are now often in the outermost periurban areas, frequently along the new main roads. Located far away from the city, the pre-existing small rural villages have grown in size and have often turned into "periurban craft villages" [55] (p. 227): the traditional farming households have obtained a significant source of additional income from craft and semi-industrial activities. Concentrated mainly in the northern region of Vietnam, these craft villages have attracted up to 11 million people in recent decades [58], resulting in significant levels of urban diffusion (as evidenced also by landscape metrics) and environmental pollution.

For accurate planning, it can be useful to also understand the dynamics and spatial patterns of urban expansion on a topographic scale, as provided by Landsat images. This is even more important if urban planners are unable to play an active role in shaping the development of the territory due to speculative interests on the part of stakeholders [59]. The Hanoi Master Plan to 2030, implemented in 2011, highlights not only the opportunities, but also the challenges involved in bringing about sustainable development of these metropolitan areas [14] (p. 77): for the time being, for example, the idea of creating a green corridor to the west of the urban core, "a green belt that protects productive farmlands, flood management areas, natural areas, craft and trade villages, and historic relics", seems to have been shelved. By focusing on the recent changes occurring in Hanoi and its surroundings, our results have mapped out the current situation, while being fully aware of the compelling need for further research that can constantly monitor the future urban growth in this beautiful ancient city.

**Funding:** This research received no external funding.

**Conflicts of Interest:** The authors declare no conflict of interest.

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
