# Peer review of "Rural–Urban Transition of Hanoi (Vietnam): Using Landsat Imagery to Map Its Recent Peri-Urbanization"

_ijgi, doi:10.3390/ijgi9110669_

Round 1

Reviewer 1 Report

This article present a good exercise of GIS analysys of the land use/ landcover of an interesting urban area of Vietnam. It seems an study case from the doctoral disserattion of the author, focused on the theme of the urban transition from socialism to capitalism, that we analyzed some years ago for the case of Bosnian Sarajevo.

In my opinion the study needs some fieldwork in order to complete itsremote vision.

But, the main problem is the theoretical one. Today the traditional contradiction between city and country needs new approaches, in the way of the Urban Theory Lab of the Harvard University. Urbanization can't be understand anymore from the only morphological point of view. The article only point de quantitative approach of the UNO and concludes that the third ring of Hanoi is not anymore rural, with no theoretical explanation.

Author Response

Dear reviewer,

thank your suggestions.

Unfortunately, I'm not as young as a PhD student, but your comments were very interesting to me. They helped me to better define the remote vision of the paper, but also the overall theoretical framework.

I considered them and I tried to answer you.

In attachment, the revised paper that follows your comments. To help you, I highlighted the revisions in cyan color.

Comments:

1. In my opinion the study needs some fieldwork in order to complete its remote vision.

Answer. In regard to this question I have added information on: a) the dataset: I have specified the reason why I have collected autumn images (LN 200-202); b) the data processing: I better specify how I decided the total number of GCPs and how I used the reference maps in the accuracy assessment procedure (LN 227-231); definition of rings: the reason why I chose the 10 Km buffer zone in the GIS-based buffer analysis (LN 246-249); the results: I better define the results of the accuracy assessment in regard to the different LULC (LN 320-324).

 2.But, the main problem is the theoretical one. Today the traditional contradiction between city and country needs new approaches, in the way of the Urban Theory Lab of the Harvard University. Urbanization can't be understand anymore from the only morphological point of view. The article only point de quantitative approach of the UNO and concludes that the third ring of Hanoi is not anymore rural, with no theoretical explanation.

Answer. Regarding this comment, I have included just few hints (also due to the editorial standards) within the text in the introduction (LN 80-86; footnote 1) and in the conclusions (LN 588-590).

Reviewer 2 Report

No particular comment.

Suggestions for future research:

  • test and use other spatial indices, specific for the analysis of the expansion of urban areas and rural-urban dynamics;
  • investigation of the socio-economic implications of LULC transformations;
  • assessment of the loss of agricultural land and natural environments and the degradation of the landscape in the areas surrounding the city of Hanoi

Author Response

Dear reviewer,

I would to thank for your suggestions: they are very interesting and I'll certainly consider them in the next step of this research or other studies about these topics.

Giovanni

Reviewer 3 Report

Dear Authors,

The study is potentially interesting and written well. But I would like to suggest the following list of revisions/comments to the manuscript to improve the quality of the manuscript

Revisions/comments

  1. LN 8-10: It is better to rewrite the long sentence into smaller sentences to improve readability. I can see this issue in several places of the manuscript and suggestions to improve the readability of the manuscript
  2. LN 18-20: It is better to specify the spatial indexes exactly rather than saying some
  3. LN 77-78: The sentence is not clear. Put the dot at the right place
  4. Section 2.1: Better to describe the geographical setting of the study area by using the coordinate system used
  5. Section 2.1: How about the city center? Better to describe the city center and it is geographical attributes
  6. Figure 1: Better to show where Vietnam located in the world map as an overview map
  7. Figure 1: Better to put north arrow and scale on the overview maps
  8. Figure 1: Better to caption the coordinate system used for the mapping
  9. Section 2.2: How about cloud coverage of the datasets? It is better to mention the cloud coverage of the datasets
  10. Section 2.3: Are 150 points enough for the accuracy assessment? Better to cite some previous studies to justify this number
  11. Section 2.3: How about other methods such as different color composites to evaluate GCPs? I think it is better to mention more details about the selection of the GCPs those are used for the accuracy assessment
  12. Section 2.4: Better to justify the selection of the rings. You can cite some previous studies with this method or geographical setting
  13. Figure 2: Better to have a north arrow
  14. Section 2.4.1: Better to justly the use of the index RUE in this kind of a study
  15. Section 2.4.2: Better to justly the use of the indexes in this kind of a study
  16. Table 2: Is there any formulas to calculate these indexes. If so better specify those formulas
  17. Section 3.1: How about the accuracy with respect to the LULC classes? Better to highlight more details about the accuracy assessment
  18. Figure 6,7: Better to have a north arrow
  19. Section 3: Better to further justify your results especially urban expansion with some other secondary statistical data sources such as population data, number of vehicles, industries, etc…
  20. The conclusion is too long. Suggest to highlight the main findings of the study with their statistical significant

The study is interesting and written well. Even there are few things to correct I would like to suggest a minor revision

Author Response

Dear reviewer,

thank your suggestions. I considered them and I tried to answer you.
In attachment, the revised paper following your comments.

To help you I highlighted the revisions in green.

  1. LN 8-10: It is better to rewrite the long sentence into smaller sentences to improve readability. I can see this issue in several places of the manuscript and suggestions to improve the readability of the manuscript   Answer: done. In detail: LN 8-10; LN 71-74; LN 110-113; LN 141-143; LN 274-277; LN 402-405; LN 516-520; LN 578-581.
  2. LN 18-20: It is better to specify the spatial indexes exactly rather than saying some. Answer: done; sentence modified (LN 18-21).
  3. LN 77-78: The sentence is not clear. Put the dot at the right place. Answer: done; sentence modified (LN 79-80).
  4. Section 2.1: Better to describe the geographical setting of the study area by using the coordinate system used. Answer: done (LN 141-142) and (LN 181)
  5. Section 2.1: How about the city center? Better to describe the city center and it is geographical attributes. Answer: done (LN 160-163)
  6. Figure 1: Better to show where Vietnam located in the world map as an overview map. Answer: In general, I hope that Vietnam's position in the world is well known. Anyway, in the map (a) of figure 1 I have placed the neighboring countries.
  7. Figure 1: Better to put north arrow and scale on the overview maps. Answer: overlaying a lot of information on the map, I think it can create confusion. Besides, the north arrow of the map (c) of figure 1 give the right direction also of the other maps [(a) and (b)].
  8. Figure 1: Better to caption the coordinate system used for the mapping. Answer: done (LN: 176)
  9. Section 2.2: How about cloud coverage of the datasets? It is better to mention the cloud coverage of the datasets. Answer: done (line 193-195 and new column in the table 1)
  10. Section 2.3: Are 150 points enough for the accuracy assessment? Better to cite some previous studies to justify this number. Answer: done (line 219-220)
  11. Section 2.3: How about other methods such as different color composites to evaluate GCPs? I think it is better to mention more details about the selection of the GCPs those are used for the accuracy assessment. Answer: How I specified in the answer to the your comment n. 10 I used a stratified accuracy assessment with larger number for larger classes (as suggested in bibliography). I don't think Color composite is important: as reference to judge my classification I used Google Earth images (as frequently in bibliography) and old maps, so I can’t modified their color composition. However, I have added some information about the procedure with Google Earth images (LN 221-223)
  12. Section 2.4: Better to justify the selection of the rings. You can cite some previous studies with this method or geographical setting. Answer: done (LN 231-233 and 238-241)
  13. Figure 2: Better to have a north arrow. Answer: done
  14. Section 2.4.1: Better to justly the use of the index RUE in this kind of a study. Answer: done (LN 259-260)
  15. Section 2.4.2: Better to justly the use of the indexes in this kind of a study. Answer: done (LN 268-269)
  16. Table 2: Is there any formulas to calculate these indexes. If so better specify those formulas. Answer: I think these reader are quite simple to understand (they are total number, or simple sum or average area or, again, simple proportion). Anyway, I give a short description of them in column 2 of table 2. Furthermore, I pointed out the reference [42] where the reader can find these.
  17. Section 3.1: How about the accuracy with respect to the LULC classes? Better to highlight more details about the accuracy assessmen. Answer: done (LN 313-317)
  18. Figure 6,7: Better to have a north arrow. Answer: done (for fig 8 and 9, too).
  19. Section 3: Better to further justify your results especially urban expansion with some other secondary statistical data sources such as population data, number of vehicles, industries, etc… Answer: your suggestion is very interesting, but the main aim of the paper is to detail and update the changes of LULC in Hanoi and its surrounding. I have collected references to justify my comments in section 3. Have other data could be very important in a future work about a detailed focus on the periurbanization, mainly in the far periurban area.
  20. The conclusion is too long. Suggest to highlight the main findings of the study with their statistical significant. Answer: I think the conclusions are proportional to the length of the paper. There are about 4,000 (with spaces) in which I have included the main and most significant statistics of the study.

Reviewer 4 Report

The article "Rural-urban transition of Hanoi (Vietnam): using Landsat imagery to map its recent peri-urbanization" presents a interesting paper related to rural-urban dynamics of the Hanoi city. Author use Landsat time series images for LU classification and some spatial indexes to better show urban dynamics. The title was formulated correctly and refers to the content of the article. The content of the article is clear, readable and correctly divided. Paper is well structure. The background is well defined. Aims, workflow and result are clear. On the other side, classification method, spatial indices and presentation of results have already been seen in various studies and papers that have been referenced.

I suggest to author to try to better point out what is new (novelity) in this paper.

Beside that suggestions, my comments are:

  • Line 77-78 Sentence is unclear
  • Line 190 - Table 1, Please explain why you are using autumn images, why not spring images. Especially if we keep in mind that the classification of the urban area may have been influenced by vegetation, and in the spring the vegetation is less lush, please provide additional explanation on the choice of imaging dates
  • Line 198 - 199 " b) the masking of mosaicked scenes, using the polygon vectors of the study area". What is masking, do you mean subset?
  • Line 200 " rule classifier: Maximum Likelihood Classification" why this rule classifier, or why only this, because there are a lot of classifiers that gave better accuracy results - explain
  • Line 292 Table 2, you mean Table 3?
  • Line 342-343 "We shall now", "we came up with" - there is only one author of this paper, but in your sentences is always we
  • Line 470 "an average of 870 hectars" - how can we compare this area with information about LULC and Cultivated areas; D: Discontinuous built-up areas; U: Urban areas; I: Industries and commercial complexes
  • Reference 22 - bad DOI link
  • References - please use same style for doi links

After implementing these revisions, as far as I'm concerned the article can be accepted.

Author Response

Dear reviewer,

thank your suggestions. I considered them and I tried to answer you.
In attachment, the revised paper following your comments.

Highlighted in yellow: new (or modified) sentences; in red: deleted sentences.

In detail:

  1. I suggest to author to try to better point out what is new (novelity) in this paper.                                                                                                    Answer: I tried to point out the noverlity inserting a sentence in the introduction (line 120-122)
  2. Line 77-78 Sentence is unclear.    Answer: sentence modified (line 78-79)
  3. Line 190 - Table 1, Please explain why you are using autumn images, why not spring images. Especially if we keep in mind that the classification of the urban area may have been influenced by vegetation, and in the spring the vegetation is less lush, please provide additional explanation on the choice of imaging dates. )                                                                            Answer: In effect, your question is very appropriate. However, I collected autumn images because they were usually the only ones available without cloud coverage. I have considered your suggestion to better specify this point in the text (line 188-189).
  4. Line 198 - 199 " b) the masking of mosaicked scenes, using the polygon vectors of the study area". What is masking, do you mean subset?    Answer: Really, subsetting and masking procedures are quite similar. Using the mask tool, I clip the raster data following the boundary of the districts within a 30 km from the center, so the resulting area is not perfectly a circle, but something slightly different. In the text, I have replaced the term ‘masking’ with ‘clipping’, so hopefully it is clearer (line 203).
  5. Line 200 " rule classifier: Maximum Likelihood Classification" why this rule classifier, or why only this, because there are a lot of classifiers that gave better accuracy results – explain.                                                    Answer: During the classification step of the procedure, I tried the different classification methods of ENVI 5.0 such as, for instance, unsupervised or spectral angle mapper classification, but the results were not as good. I could also verify the same problem using other rule classifier in the supervised procedure, like minimum distance classification or parallelepiped classification.  For these reasons, I have chosen this rule.
  6. Line 292 Table 2, you mean Table 3?  Answer:  Yes. I changed it in the text (line 296).
  7. Line 342-343 "We shall now", "we came up with" - there is only one author of this paper, but in your sentences is always we.                            Answer:  In effect, your remark is formally correct. I verified that the use of “we” occured thirty times in the paper. However, in the accademic writing and in the scientific papers I cannot use the pronoun “I”, while the pronoun “we” is usually tolerated to eliminate passive voice constructions. Such constructions would risk making the text a little too complicated. For this reason, at the moment I have chosen to keep the use of  “we” as a pronoun, but if you prefer I can change.
  8. Line 470 "an average of 870 hectars" - how can we compare this area with information about LULC and Cultivated areas; D: Discontinuous built-up areas; U: Urban areas; I: Industries and commercial complexes.         Answer: Since this sentence, if not contextualized as you indicated, makes little sense, I decided to delete it (in red, line 473-475)
  9. Reference 22 - bad DOI link.   Answer: I corrected it
  10. References - please use same style for doi links  Answer: Ok

Round 2

Reviewer 4 Report

The changes made in the paper and the responses to my comments have made a significant improvement in the work. The paper can be accepted in this form.